# Psychological Distress of University Workers during COVID-19 Pandemic in Brazil

**DOI:** 10.3390/ijerph17228520

**Published:** 2020-11-17

**Authors:** Fernanda Barcellos Serralta, Murilo Ricardo Zibetti, Chris Evans

**Affiliations:** 1Post Graduate Program in Psychology, Universidade do Vale do Rio dos Sinos—UNISINOS, São Leopoldo 93022-970, Brazil; murilozibetti@unisinos.br; 2Department of Psychology, The University of Sheffield, Sheffield S1 1HD, UK; chris@psyctc.org

**Keywords:** COVID-19, pandemics, quarantine, psychological distress, cross-sectional studies

## Abstract

The study aimed to explore mental distress during COVID-19 quarantine in a sample of university workers in Brazil. The survey included sets of questions about demographics, health, and support, an open question about major concerns, and the Clinical Outcome Routine Evaluation (CORE-OM), a measure of mental distress. A total of 407 professionals, mean age 40, SD 11.2, fulfilling social distancing (99%) participated in the study. Participants were mostly female (67.8%) and married (64.8%). Using the Consensual Qualitative Research process for simple qualitative data (CQR-M), the main areas of concern were grouped into six domains, as follows: work, health, isolation, personal life and routine, social environment, and future. Many responses were multiple. They form categories indicating specific concerns within these domains. Quantitative data were analyzed by identifying the simple effects of potential predictors of mental distress. The results indicated medium effects of help with household chores (η^2^ = 0.06, 95% confidence interval (CI) (0.022–0.095)), psychiatric treatment (η^2^ = 0.06, CI (0.030–0.110)), age (η^2^ = 0.12, CI (0.070–0.170)), and physical exercise (η^2^ = 0.12, CI (0.079–0.180)). Having someone available to listen was the only variable with a large effect associated with reduced mental suffering (η^2^ = 0.18; CI (0.118–0.227)). Psychological experiences of the pandemic are multifaceted and complex. Thus, substantially larger surveys, with both quantitative and qualitative components, are needed.

## 1. Introduction

On 11th March 2020, the WHO announced the pandemic status of COVID-19 (CV-19) infection. Worldwide social distancing measures were adopted to prevent virus exposure. According to the rate of transmission, policies varied between limitations on social gatherings, through to “lockdown”: extreme restrictions on interpersonal contact. This completely new situation has consequences yet to be measured and understood by health professionals and researchers. Previous studies carried out in situations similar or analogous to the current pandemic show these have a deep and wide impact on the mental health [1]. Such impacts include, among others, the development of clinical conditions in hitherto healthy people and the worsening of pre-existing conditions. In addition, these conditions tend to persist in the long term even after the event that caused the crisis ceased [2].

It is well known that, in epidemics, substantial mental health problems are experienced both by those directly affected by an infectious disease but also by many who are not infected. The pandemic fear creates increased levels of psychological distress due to fear of contagion, social isolation, economic loss, changes in the family and work environment, and among other factors, causes feelings of abandonment and hopelessness, producing diverse conditions, including anxiety disorders, depression, and suicide [3]. 

Psychological distress is as an emotional state of suffering comprising symptoms of depression, anxiety, stress that may be tied in with somatic complains and can be seen as a response to exposure to stressful events that threatens either or both physical and mental health [4]. The literature underlines that pandemics are disruptive to mental health and produce negative affective reactions. It appears that some people are more vulnerable than others are, and may experience intense psychological distress, which can lead to serious behavioral or psychological problems, especially when unnoticed and therefore untreated. 

Surveys carried out in China indicate that most of the population perceived a moderate or severe impact of the pandemic on their mental health [5]. Approximately one third of the people in the territory show signs of peritraumatic psychological suffering [6]. In Italy, a survey indicated that a quarter of people had levels of anxiety and depression and almost half had a sleep problem [7]. A large study with probabilistic sampling of the UK population indicated that, comparing before and during the lockdown, there was an increase in the overall mental distress, with the prevalence of clinical levels reaching 27.3% of the participants. [8]. In the USA, a national representative study conducted during an acute period of COVID-19 spread found increased symptoms of anxiety and depression over time, and estimated that, in the most affected states, each day past since March 10 implied an 11% greater chance of moving to the next category of distress (e.g., moving from mild to moderate symptoms) [9]. These kind of effects of the pandemic have been reported by several other countries, including in Latin America, where contagion started later than in Europe and North America [10,11,12]. 

As happened in other countries, in Brazil, empirical studies on mental health during early stages of the pandemic have developed rapidly. An online survey [13] carried out on a community sample of 799 participants from the state of Rio Grande do Sul identified that being female, younger, and the presence of previous disorders were predictors of mental disorder symptoms during the pandemic. In addition, it was observed that effects of the current context such as being in the risk group, losing income and being exposed to negative news increase the risks to mental health [13]. A large survey study of 45,000 people found that more than 40% of them reported anxiety, or sadness and depression, or difficulty sleeping. Young adults, women, and people with previous depression were the most distressed [14].

Despite the efforts of researchers, very little is currently known about the psychological impacts of COVID-19 and of transmission reducing public health measures across different countries and regions of the world. Apart from being a completely new threat, psychological reactions can vary across groups and cultures [4]. Therefore, at this stage of knowledge, it is important to open research methodology and not prejudge impacts. Surveys that rely exclusively on quantitative data are inevitably “researcher down”, as they do not allow any subjective, qualitative responding. The bottom-up approach is suitable to investigate specificities of groups or segments of population regarding the experience during social isolation, therefore providing more in-depth information to generate both preventive and rehabilitation actions in mental health. One of the few qualitative studies conducted in Brazil on this topic aimed to access the social representations about the coronavirus pandemic and its treatment. The representations of the pandemic were divided into two thematic axes: (a) concept, contamination, and prevention of COVID-19; (b) psycho-affective and social implications of the pandemic. The latter involves concerns about virus dissemination and its psychosocial and affective implications. Regarding treatment, the main themes related to the search for a cure (institutional responses to contain and develop treatments) and the economic and social difficulties in accessing treatment [15]. 

To plan psychological interventions targeting promotion of, and/or restoration of, mental health we need studies aiming at identifying populations at risk in terms of psychological distress, as well as studies focusing how people experience the life-threating situation. Worldwide, cohort studies identified psychological distress in general population [4,5,6,7,8,9,10,11,12]. Some surveys targeted the mental health of a specific population, such as healthcare workers, students, vulnerable populations, and workers of impacted economic areas [16]. However other supposedly vulnerable groups were largely neglected. This is the case of university workers. 

COVID-19 is exerting a significant and negative impact on education and in mental health on students [17], teachers and other workers [18,19]. In Brazil, as in many other countries, while restaurants, shops, factories and almost all services were closed at early and rapidly increasing stages of pandemic, most universities closed for only a few days to reopen almost immediately to deliver online classes and other remote services to students and community. The need of university employees having to adapt to this radical work changing scenario coexists with their need to deal with all other social, psychological, and physical threats imposed by pandemic outbreak. One of the few studies focusing psychological distress in a university community, including its staff, found increased sleep problems in administration workers during lockdown in Italy [18]. Another study conducted during lockdown in Spain found that university workers, in comparison to students, reported more concerns about health, social and economic issues [19]. In both studies, moderate to severe symptoms of common mental disorders, like anxiety and depression, were prevalent in approximately a third of the sample, but higher among students [18,19]. Taken together, these results reinforce the premise that different subgroups of the population are affected differently by this stressful event.

The study reported here is a hybrid, quantitative and qualitative, survey aiming to map the psychological distress and the conditions associated with it during the quarantine imposed by the COVID-19 pandemic in all level staff from a Brazilian private university. The study is pragmatic and derived from a larger research intervention protocol intended to provide psychological care and psychoeducation to alleviate distress and promote mental health among university community. The original protocol includes repeated measurement and assessment of intervention outcomes, to be reported elsewhere. We believe that this kind of mental health applied research can be useful to inform public health research and programs. 

## 2. Materials and Methods 

This is a cross-sectional quantitative and qualitative study inquiring about mental health conditions from university staff about two months after beginning of social distancing recommendations in Brazil. At that time, rate of infection in the country was rapidly increasing and in northern regions health system was collapsing. In south of Brazil, where data were collected, the virus was spreading with much less speed, supposedly because of the early social distancing measures adopted locally. 

### 2.1. Participants

The population eligible to participate were employees and indirect workers (*N* ≅ 1850), approximately 55% female, most with full time job contract) from Universidade do Vale do Rio dos Sinos, a large private university located in southern region of Brazil. There were 407 respondents, one of whom was declared as transgender (omitted from analyses of gender as a predictor). Of the remaining, 276 (68%) were women. Age ranged from 19 to 71 with mean (M) 40.0 and median (Md) 38.

### 2.2. Instruments

A single questionnaire developed by researchers containing general information on four sets of variables: non COVID-19 demographics, COVID-19 related demographics, self-care/health activities, and general and professional support. Most questions were structured with simple or multiple choices A single open-ended question was included asking about the main stressors that the participants were currently experiencing. This question was intended to capture subjective aspects and areas of psychological distress related to pandemic. In addition, the Clinical Outcome Routine Evaluation—Outcome Monitoring (CORE-OM) [20,21,22] was used (for a summarized description of these variables, see Table 1). 

The CORE-OM, is a self-report scale developed in UK during the 1990s to support monitoring of change and outcomes in routine psychological and health interventions. It is free of charge and widely used in many countries and languages, giving a brief but broad coverage of distress and dysfunction. The questionnaire contains 34 items that assess four domains: subjective well-being (4 items); problems and symptoms (12 items); life functionality (12 items) and risk for yourself and others (6 items). These items are answered on a five-point scale, and range from “never” to “always or almost always”. The measure has six scores reflecting items from those four domains plus the total score and the “non-risk” (NR) score of the 28 non-risk items. The scores reflect content domains not population dimensions or factors. Most studies of the internal structure show the risk and NR items to have quite low correlation with each other and clinically the risk items are recommended to be treated as “flags” rather than making up a psychometrically strong score while the NR score is a clear measure of psychological distress. Though we used the risk items to identify individuals at particular risk, this report uses the NR score as our main response variable.

The Brazilian Portuguese version of CORE-OM [23] was developed following the guidelines of the CORE System Trust (www.coresystemtrust.org.uk/cst-translation-policy). Studies of the psychometric properties of this version in the Brazilian population have not yet been reported but a well-powered initial study is currently being prepared for submission by the authors together with other colleagues and shows reassuring psychometric properties. Although translations and geographically different populations result in different psychometric properties from those of the original measure in the original population, a psychometric exploration of the CORE-OM in many translations has shown only minor differences in psychometrics from the findings of the original exploration. A recent study conducted in Ecuador with the CORE-OM suggest the Spanish translated measure is valid, reliable, and suitable for the Latin American population [24]. The initial study, in UK with a clinical sample (*n* = 890) and a non-clinical sample (*n* = 1106), found a good index of internal consistency (Cronbach’s α = 0.94) for the NR score (in both samples) and test-retest correlations in a student sample was 0.91. Convergent validity with several symptom scales showed correlations between moderate and high (R ranging between 0.56 and 0.85) discrimination between clinical and non-clinical samples also good with Cohen d 1.7 [21]. In this dataset Cronbach alpha was 0.947 with 95% confidence interval (CI) from 0.938 to 0.954.

### 2.3. Procedures

Data collection occurred in May 2020. An invitation to participate in the survey together with a link to the form was sent by email to all individuals in the university official list of employees and workers. The forms remained open to responses for ten days. This timeframe was estimated as long enough to offer reasonable time for workers to voluntarily engage in research, but also short enough to prevent major variation in external conditions (e.g., infection rates and social distancing policies in the region) and to allow for the planning and implementation of the already mentioned psychological interventions that were part of the broader study protocol in a short period, i.e., less than one month.

### 2.4. Data Analysis 

The epistemological position is pragmatic and contextual: we believe the evidential value of the data is defined by its actual or potential utility locally or more widely and located within the dual contexts of psychological and public health research. We used descriptive analysis to summarise quantitative socio-demographic, general and mental health, and self-care data. This epistemological stance looks only for cautious extrapolation to other settings and populations on the principle that new data is valuable, and particularly when there is little existing data addressing the issues, it is a rather different stance from empirical positivist epistemology often linked to an aspiration for large and clearly representative samples that might give strong generalizability.

The open-ended question was analyzed using Consensual Qualitative Research for simple qualitative data (CQR-M) [25]. The Consensual Qualitative Research (CQR) [26] from which the CQR-M was built is an eclectic qualitative method that combines elements of grounded theory, constructivism, phenomenology, and post-positivism to explore or understand a phenomenon experienced by groups of individuals. The method uses consensus between judges in order to capture multiple viewpoints about the data (a form of triangulation). The team of raters were six psychology students that were trained to CQR-M by the first author. Since raters were inexperienced in qualitative analysis, the first author audited all steps. First, pairs of independent raters examined one third of answers each to list different domains, i.e., broad topics to group data. All reviewed this list, and final domains were established in a consensual meeting with all raters and auditor. Then, raters returned to material and grouped ideas into similar categories. In another consensus meeting with all raters and the auditor, the final categories were defined. Subsequently, the auditor reviewed all categories. Then, we verified the frequency of each category occurrence; and finally, made small adjustments in order to prevent having categories with very small frequency (<1%). Miscellaneous irrelevant data were excluded (e.g., general comments about the survey). 

The quantitative analytic approach was descriptive and exploratory: we recognise the data are from a subset of the employees of one university and wider generalization to the whole employee group, or wider, must be extremely cautious. The aim was to identify any strong relationships between other variables (“predictors”) and psychological distress/dysfunction as measured by the CORE-OM NR score (“dependent”). Null hypothesis tests of association with a conventional alpha level of 0.05 were used to filter out associations and identify those of potential interest. For tests on continuous variables, bootstrap p values were used (1000 bootstrap replications). Clearly, although many associations explored the likelihood of some false positively significant associations/effects as high, as the testing was only filtering in effects for further comment and as all effects of interest were then given effect sizes to allow comparison of their possible impacts on the response variables, the costs of this approach seemed lower than those using any of the possible multiple tests “corrections” across so many and diverse associations. 

Moreover, 95% confidence intervals around observed sample statistics are used as these convey the precision of estimation of the population parameters given the sample size (but the estimation will always be biased by selective non-participation so CIs themselves must be interpreted cautiously). The 95% CIs for effect sizes were parametric, all others were bootstrap CIs based on 1000 bootstrap samples. To avoid the analytic realms being treated as unconnected, and to see what information can add to the other, qualitative domain codings were also treated as quantitative variables hence the CORE-OM NR score was explored for associations with the following.

Whether a free response was coded into one (or more) of the seven domains emerging from the qualitative analysis (see details below).Non-COVID-19 demographics: gender, age, marital/relationship status, household size, number of children.COVID-19 related demographics: member of an essential work group, having a member of the household in such a work group, being personally at high risk if infected, having been infected, having someone in the household who had been infected.Self-care/health activities per week: days alcohol consumed, days unprocessed food consumed, days exercised, days had some relaxation.General and professional psychological support: perception of help being available, perception of having listeners available, experience of counselling, experience of psychiatric support.

We recognised that there would be three major challenges to any simple analysis of the relationships between these predictors and the NR score response variable: associations between predictors, non-linear relationships between continuous predictors and response, and statistically significant interactions between predictors in their effects on dependents. Wherever these applied we have flagged this by referring to the predictive relationship as “complex”. With these issues in mind the quantitative analyses started with an exploration of associations between the major predictor variables to identify any strong and statistically significant associations. This was followed by a fairly exhaustive exploration of univariate predictor/response relationships with some exploration of linearity (for predictors with at least four ordered categories). Finally, bivariate interactions were explored for adding gender and age with each of the other predictors. The full exploration is available by request to the authors. The most clear and interesting relationships are reported next with the strength of the relationships indexed with η^2^ (eta squared) with its 95% parametric CI, the effect sizes (Figure 4) were crudely classified as small, medium, and large.

### 2.5. Ethical Considerations

The Research Ethics Committee from university of origin approved the study protocol (CAAE: 31225520.0.0000.5344). All participants signed the online consent form for the study confirming free and informed consent. Respondents´ private information was protected at all stages. To ensure confidentiality and reliability, a personal alphanumeric code was generated and used as identification. 

## 3. Results

### 3.1. Participants Socio-Demographic and Psychosocial Characteristics

Participants were 407 professionals, majority female (67.8%), married (64.8%). Mean age was 40 (SD = 11.12). Most were working in technical-administrative functions (50.9%), remotely, at home (85.7%). The vast majority, 264 (64.7%) of participants reported working hours as 40, other responses ranged from 4 to 60 (M = 34.27; Md = 40).

As with the number of weekly hours of work, cumulative days in social distancing had a very strong single peak with 196 participants (48.2%) saying 60 days. The vast majority of sample (98.8%) declared compliance with official social distancing recommendations and were going out of the home only once or twice a week for essential tasks (82.6%). Respondents often or almost always had support from others to help them with household chores (68.6%) and to share their problems and concerns (70.5%). 

### 3.2. General Health Status and Behaviors

21.4% reported having previous health conditions associated with high risk to COVID-19 complications. None of the participants had tested either positive or negative for the virus, although 42 (10.3%) reported having experienced mild or moderate COVID-19-like symptoms. 

A significant portion of the respondents were under mental health treatments when pandemic started, 21.6% assisted by psychologists and 10.1% by psychiatrists. In addition, some others sought mental health assistance after the pandemic either with psychologist (1.7%) and/or with psychiatrist (1.7%). 

Regular alcohol consumption was prevalent in 64.9% of participants, with 17.1% having some alcohol at least three times a week. Healthy habits were also highly prevalent: the majority ate homemade meals daily (64.4%), exercised at weekly basis (68.3%) and were practicing hobbies that promote relaxation (e.g., meditating) weekly (61.1%).

### 3.3. Major Concerns–Qualitative Data

Only 12 (2.9%) participants did not offer any qualitative comment. The participants who did (*n* = 395) gave 487 responses that were classified into six domains: work (*n* = 111, 22.79%), health (*n* = 84, 17.25%), isolation (*n* = 73, 14.99%), personal life and routine (*n* = 90, 18.48%), social environment (*n* = 48, 9,86%), and future (*n* = 67, 13.76%). A seventh small set (*n* = 14, 2.87%) included responses that explicitly disavowed being stressed by COVID-19. Only 31participants (7.6% of the 407) gave responses which did not clearly fit into the domains and most of those were comments on the survey, these responses were excluded as unrelated to aims of the study. Most participants had the content of their response allocated to only one of the categories (N = 269; 66.1%). In addition, some participants provided more complex responses whose content covered two (N = 76; 18.7%), three (N = 14; 3.4%) or four (N = 3; 0.7%) domains. Categories falling in these domains are presented in Table 2, together with their frequencies in the domains. 

### 3.4. CORE-OM NR-Quantitative Data

As noted above, mental distress was measured with the Non-risk (NR) scale of CORE-OM. Scores ranged from 0 to 3.39 (the possible score limits are 0 and 4) with mean (M) 1.05 (CI (0.989–1.12), median (Md) 0.893 (CI (0.821–1.00)) and standard deviation (SD) 0.658 (CI (0.610–0.708)) and the distribution showed a clear if mild positive skew (skew 0.824, CI (0.655–1.06)) typical of Mental Health scores in general population samples. A minor convergent validity check of the NR score as a marker of distress denotes a very marked mean score difference between those explicitly saying they were not stressed by CV-19 (M = 0.31) versus those who declared stress within at least one of the qualitative domains (M = 1.08, difference 0.77 with 95% CI (0.649–0.890)). 

### 3.5. Notable Predictor-Response Associations

We tested the direct effects and potential predictors in the measure of mental distress (CORE-OM NR). We also considered complex effects derived from the interaction between potential predictors. We understand that the interaction between the other predictors can eventually change the interpretation of some of the results. For this reason, we present some of the most interesting associations between potential predictors, but we limited this analysis to the interaction between gender and age with the other potential predictors.

As expected, there were several statistically significant associations between potential predictors. Some of these are entirely logical (e.g., between number in the household and the number of children and those at high risk from COVID-19 being older than those who weren’t). Others were emergent findings (e.g., that those with others in the household in CV-19 essential occupations were younger than those without). 

We explored some interesting interactions. Having ever had support from a psychologist was associated with gender: 68.8% of women having, or having had, such support vs. 46.1% of the men; for psychiatric help the proportions were 36.1% vs. 23.9%. Age was not related to psychological help but there was a marginally significant tendency for those currently seeing a psychiatrist to be younger than those not seeing one. Perception of help available was not statistically significantly linked with gender or age, perception of having listeners available was not related to gender but was associated with age: increasing with age. Perception of help and of listeners available were positively correlated (R = 0.36 (95% CI 0.26–0.45)) None of the four health behaviours showed gender differences and only getting exercise showed a relationship with age: a curvilinear relationship with lowest rate of exercise at age 30 and a clear rise either side of that age. There was no statistically significant gender effect (bootstrap *p* = 0.12) but a strong relationship with age with a broadly linear decrease in scores with age (η^2^ =0.122, CI (0.077–0.173)), as shown in Figure 1. The age effect was independent of gender. 

There was a statistically significant effect of marital/relationship status with partnered participants having higher NR scores than single participants: mean difference −0.15 (CI (−0.28–0.01)). There was a complex and statistically significant effect of gender and household size on NR score (Figure 2): neither simple effect of household size or gender (as noted above) was statistically significant but there was a statistically significant interaction (*p* = 0.02) between the two predictors with women’s mean NR scores tending to increase with household size while men’s scores were not associated with household size. There was no effect of number of children. 

It is noteworthy that higher NR scores were detected in people experiencing COVID-19 like symptoms compared with those who were not (η^2^ = 0.015; CI (0.002–0.04)). A marked effect was found with higher NR scores was also found if another person in the household was experiencing COVID-19 symptoms (η^2^= 0.028; CI (0.007–0.059)). It is worth remembering that, until the moment of the research, these people did not have a diagnosis confirmed by laboratory tests. By contrast, surprisingly, being at high risk from COVID-19 was associated with lower mean NR scores with mean difference 0.19 (CI (0.03–0.35)). And there was no statistically significant effect of being an essential worker (η^2^ = 0.00; CI (0.0–0.0)) or living with someone that is essential worker (η^2^ = 0.01 CI (0.00–0.031)).

Another group of predictors were self-care/health activities per week: days alcohol consumed (η^2^ = 0.022; CI (0.01–0.046)), days unprocessed food consumed (η^2^ = 0.046; CI (0.014–0.078)), days exercised (η^2^ = 0.13; CI (0.079–0.178)) and days relaxing (η^2^ = 0.37; CI (0.009–0.066)), the first two showing no relationship with NR score and the last two showing a statistically significant fall in NR score with increased numbers of days per week of exercise or relaxation.

The last group of predictors were perception of receiving general and professional psychological support (Figure 3). About general support, somewhat surprisingly the perception of help having been available was statistically significantly associated with NR score but with a strongly curvilinear relationship, not the simple trend we expected (see Figure 3C). The curvilinear relationship had a medium effect size (Figure 4). On the other hand, perception that listeners were available was very linearly linked with NR scores (η^2^ = 0.176; CI (0.118–0.227)). 

Regarding professional help, when we compare the group that never experienced counseling (M = 0.912; CI (0.817–1.0)), to people that did it in the past (M = 1.13; CI (1.04–1.23)) and to people that are experiencing now (M = 1,17; CI (0.99–1.37)) we can see that this variable is related to mental distress as measured by NR score (η^2^ = 0.029; CI (0.006–0.059)). The same effect is observed comparing groups that never (M = 0.943; CI (0.88–1.01)). In the past (M = 1.230; CI (1.09–1.37)) and now (M = 1.43; CI (1.16–1.72)) experience in psychiatric treatment (η^2^ = 0.065; CI (0.03–0.105)).

Figure 4 shows the effect sizes (ES) for the various predictor variables as predictors of CORE-OM NR score. Rules of thumb levels for the ES are shown by the horizontal reference lines and the vertical lines are the parametric 95% confidence intervals for each predictor’s effect. The black points and CI lines are the raw effects and the predictors have been ordered by raw ES. It can be seen that ES range from zero for the participant’s work being essential up to a large effect size for rating of having listeners. The blue points are effect sizes after partialling out gender, red after partialling out age and purple after partialling out both gender and age. It can be seen that the effects of partialling out gender are small, the biggest being a slight increase in the ES for the association of days per week eating unprocessed food. By contrast, the effects of partialling out age are more marked and increase the ES for days of alcohol per week, don’t much affect some ES, e.g., gender and number of children, and they reduce other effect sizes (social status, being in a CV-19 high risk group and days of exercise per week).

## 4. Discussion

The new coronavirus pandemic is supposedly producing negative impacts on Mental Health not only because anyone can be infected and therefore have a potential traumatic experience of being severely ill, but also because of the indirect effects of social deprivation, reduction of liberty, economic losses, and higher exposure to adverse home environments [27]. The psychological distress due to this collective trauma exposure need to be understood in order to be cared for and mitigated [28]. In response to potential harms presented by COVID-19, many companies, schools, and universities stayed open and asked their employees to work remotely, at “home office”. In order to contribute for understanding on how university workers were affected by this outbreak and changes it produced in their environment we conducted a quantitative and qualitative study mapping psychological distress and conditions associated to it in staff from a provide university in southern Brazil. It is noteworthy that the vast majority of participants answered the open-ended question about current stressor or major concern. From qualitative analysis, we learned that most people must deal with major concerns related to the pandemic embracing their work, health, isolation, personal life and routine, social environment, and future. Although these results show some correspondence with the thematic axes capturing general population social representations in the pandemic in Brazil identified by Do Bú and collegues [15], the scope of our health domain is relatively narrower and most of our categories could be understood in relation to their psycho-affective and social implications of the pandemic and our findings corroborate the argument that multiple elements parallel to contagion are potentially harmful to psychological health [27]. 

The work and personal life and routine domains reflect some of the mental health vulnerabilities related to home office transition, including work overload, “digital fatigue”, and loss of boundaries between private and professional life, for example. Others have noticed that COVID-19 found most teachers less prepared to virtual, “screen-to-screen”, teaching [29,30]. Our results suggest that during pandemic the promotion of digital competence among university teachers and other employees must be accompanied by mental health promotion. Almost one third of qualitative responses indicated more than one area of vulnerability. For some, the fear of contamination co-occurs with loneliness and feelings of imprisonment due to social isolation. For others, changes in work (e.g., more demands and more online tasks and interactions) are exhausting and co-exist with sense of loss of personal time, or depressive symptoms, for example. Within each domain, many responses also fell in multiple categories (e.g., a respondent was worried about someone´s health, but also mentioned having developed insomnia). These findings show that the pandemic produces broad concerns and affects many different aspects of life and that the combination of impacts varies across individuals. Therefore, researchers and mental health professionals providing support should avoid thinking only of simple and direct relationships between factors and distress, as well as overlooking some vulnerable subgroups of population like university staff. 

From quantitative analysis, we identified that our sample had overall level of mental distress comparable to those found in community [21] and university samples [24] in other countries. Age, physical exercise routine, perceived help for household chores, history of psychiatric treatment (an indirect assessment of mental health condition) and having listeners available (social support) were predictors of mental distress during pandemic. All predictors were in the expected direction and replicate effects reported in the general population of other countries [6,7,8] and in Brazil [13,14], as discussed next. In general, our results also give a more in-depth understanding of symptoms [18] and concerns [19] previously reported by university workers in European countries.

It is noteworthy that up to the time of data collection, essentially none of the participants or their families had had a diagnosis confirmed by COVID-19 testing, reflecting the fact that he state of Rio Grande do Sul started social distancing early. However, the perception of COVID-like symptoms, both in participants and in people close to them, is linked to greater psychological distress showing in CORE-OM scores. Qualitative data show that a significant stress factor is the fear of becoming seriously ill or even dying and thereby leaving significant others helpless or, alternatively, being responsible for the contagion of loved ones. This type of fear, specific and uncontrolled, is one of the most common reactions in relation to pandemic exposures [1]. 

However, people who reported being part of a risk group had less distress, contrary to findings of others [17,26]. Although this result could be spurious, alternatively, it could reflect a tendency of people who do not consider themselves be in major biological risk to feel more deprived in quarantine. Our qualitative data reinforces that isolation is an important factor of vulnerability for mental health in the pandemic not only because of loss of personal contact with others but also due to loss of freedom to come and go and to decide on relationships. Diverging recommendations, even among health authorities, supposedly contribute to ambivalence towards social distancing measures to prevent COVID-19 infection, as we can infer from our qualitative data.

Regarding non COVID-19 demographic variables in relation to mental health, the statistically significant effect of age, and the absence of a significant gender effect, are notable. In our sample, older individuals tend to exhibit less psychological distress. A similar result was found in a national survey conducted in the USA in the early pandemic in which older age was also associated with less financial concern and less perception of risk of infection and of need for quarantine [31]. The absence of a significant gender effect, although unexpected, was also reported by others in China [32]. Our results suggest, however, that men and women may have somewhat different reactions depending on the characteristics of their environment during isolation, as having more people in the household was associated with greater suffering in women. This possibly reflects the fact that, in our culture, women traditionally assume more of the household chores and childcare, which could result in greater overload of activities in the home office situation. Our qualitative findings about the excess of domestic activities and the need to reconcile multiple activities that, although present in both genders, were more prevalent in women, corroborate this hypothesis.

In line to studies attesting gender effect on mental health services utilization [33,34], in our sample females tended to seek more professional psychological help than males. Unsurprisingly, participants that received in the past or were currently receiving professional support, specially by psychiatrist, were mores distressed than those who never sought mental health treatments. During the COVID-19 pandemic general practitioners and other health professionals should pay special attention to individuals with preexisting mental and behavioral problems, as well as looking out for emerging complaints in those with no history of such problems. 

Our questionnaire included some lifestyle behaviors because of their relevance to mental health and s cardiovascular morbidities [35] and because social isolation can prevent people from sustain previous healthy habits (e.g., exercising regularly) and can s increase risk behaviors (e.g., alcohol consumption). A Chinese survey found mixed effects of isolation on healthy behaviors, with increase in eating quality but less physical activity [36]. As expected, in our sample associations between these predictors and mental health were found, with a moderate effect of exercise routine. This is congruent with other research findings indicating that regular exercise can alleviate mental distress from by COVID-19 [32] and that eating quality food has a protective effect of on subjective well being [37]. 

In addition, our findings indicate that while receiving help with household chores is also important, having someone who listens to the participant has a greater protective effect on the mental health of both men and women. Thus, to mitigate anxiety and distress during and after this crisis, it may be wise to help people to interact and connect with each other. Recommendations of physical distancing that are deemed necessary to flatten the curve of COVID-19 should no longer be confounded with recommendations of social distancing [38]. Relationship is powerful antidote to mental distress. Health professionals should listen carefully to the patient’s interpersonal experiences and social support, thus helping more isolated people to find new ways to connect to others, either remotely or in face-to-face but physically distant interaction. Increasing the population’s access to psychological care services also seem necessary to mitigate psychological impacts of pandemic.

### Limitations and Strenghts

The study has many limitations. Data is from a single university. Replication in other similar population groups is necessary to achieve generalization. Low response rate may reflect bias in responding. The sample gender distribution showed an excess of female over male respondents as the 95% CI around our 276/406 is from 63% to 72% which doesn’t include the 55% female in the university employees generally. The small sample size gives limited power so some small to moderate population effects may well have been missed. The limited sample size has even more severe impact on the exploration of interactions between predictors in their associations with distress. Although the gender/household size interaction is a clear, the small n and consequently much smaller cell sizes that underpin detection of interactions severely limit the power to detect other like this. Therefore, it is possible, if not particularly likely, that the failure to detect an interaction may mean that two predictors may have shown non-significant effects on NR score because of a negative interaction that wasn’t strong enough to show as statistically significant. Because of the study low power and to avoid “multiple tests” inflation of apparently significant findings, we chose only to explore interactions with gender and age. However, we acknowledge that other significant and clinical meaningful interactions may well exist and should be explored in larger studies. Nevertheless, since the study was exploratory and we have conducted multiple tests, some effects that were significant may be spurious. We hope that focusing more on the effect sizes, with and without gender and age as predictors, and the reporting of CIs for the effect sizes, goes some way to mitigate this. 

Interpretations of all epidemiological study must be limited and cautious, since results are always complex and need to be interpreted carefully and with aid of cognitive science, i.e., considering cognitive bias, especially when numbers and quantities are unnecessarily dichotomised or in other ways simplified [33]. We believe that choosing a mixed methods survey strengthens this study and helps avoid oversimplified interpretations. One strength is that few studies examined mental distress during the COVID-19 pandemic in university workers and with the pressure of continuing work, but generally using distance methods, may make this a group at particular risk of psychological distress. Using a hybrid approach was essential to access the complex subjective dimension of distress in this population. Although our sample is a pragmatic sample, the complementary types of data, qualitative and quantitative, provided an accurate and broad picture of how the pandemic negatively affects people´s life and mental health, allowing the formulation of grounded-on-experience hypotheses of the impact of pandemic on the general population. 

## 5. Conclusions

With the pandemic many hitherto relatively stable aspects of people’s lives have changed radically: work, relationships, personal life, and routines. Our hybrid approach to survey the problem led to a multifaceted and complex picture of the consequences of pandemic not only for mental health (as an outcome) but also for psychological experience (as a process of existing, perceiving and giving meaning to experiences. Reading to respondent´s major concerns we found out that these changes produce not only psychological symptoms and health preoccupations but also loneliness and a sense of helplessness. Social, political, and economic insecurity contribute to the scenario. 

Our quantitative analysis showed several factors associated with psychological suffering. Of course, real relationships often bare associations between predictors of non-linear relationships with ordinal variables and with some strong interactions between predictors when their effects were considered together. It is noteworthy that the strongest effect size was for the effect of having someone in whom to confide and seek help with personal issues, which was associated with less distress, underlining the relevance of personal, intimate, and supportive relationships to psychological well-being. Therefore, current public policies outlined to promote population health should include provision of remote psychological support to isolated and more vulnerable individuals. More studies are needed to analyze the main effects of social distancing and COVID-19 crisis on mental health.

## Figures and Tables

**Figure 1 ijerph-17-08520-f001:**
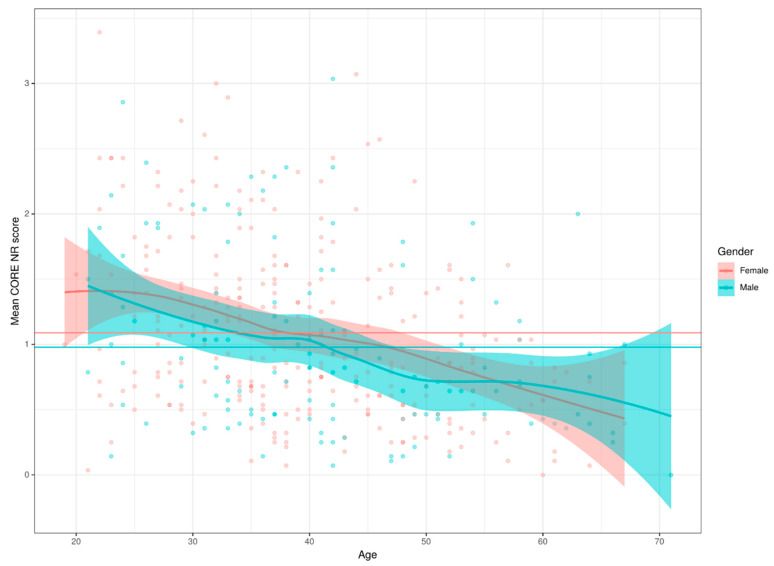
Relationship of CORE-OM NR score with age and gender.

**Figure 2 ijerph-17-08520-f002:**
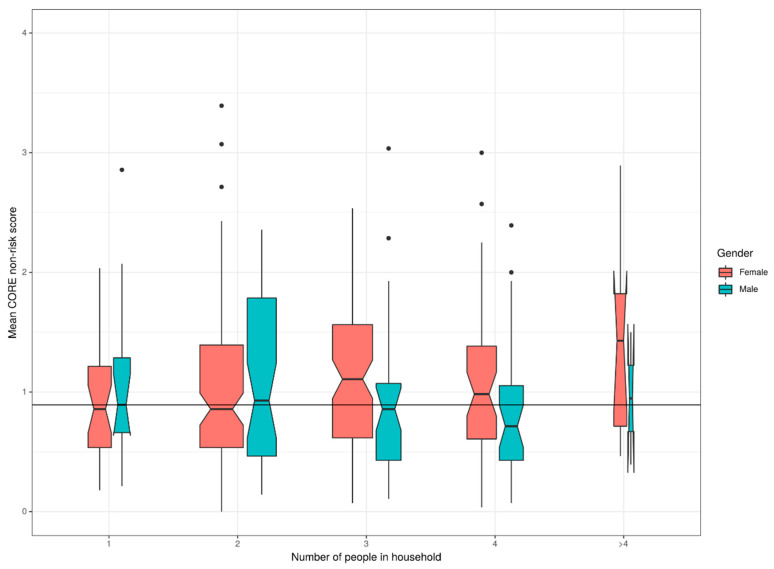
Effect of gender and household size on CORE-OM NR score.

**Figure 3 ijerph-17-08520-f003:**
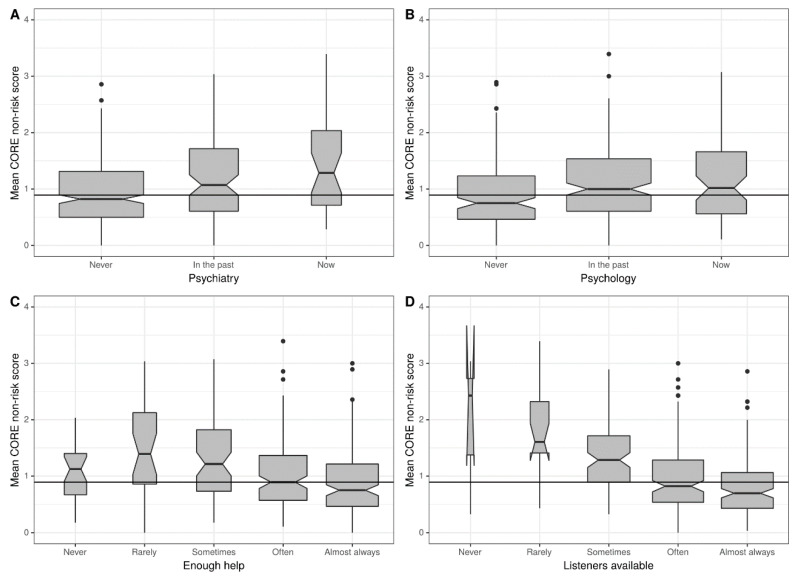
Relationship between general support and professional support with CORE-OM NR score. The four separate figures each show the relationship between mean CORE-OM non-risk score and respectively: contact with a psychiatrist (**A**), contact with professional psychological support (**B**), perception by the participant of availability of lay help (**C**) and perception of having listeners available (**D**). Box areas are proportion to cell size and notches show approximate 95% CI of the median. Horizontal reference lines are overall median scores.

**Figure 4 ijerph-17-08520-f004:**
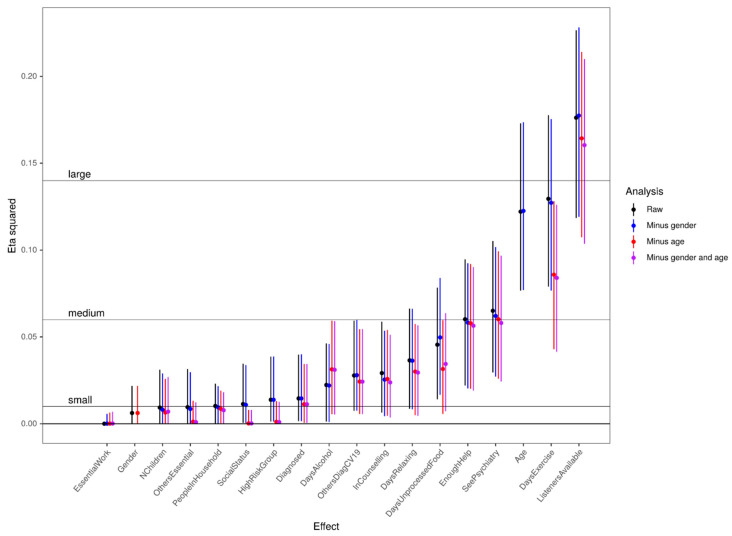
Effects sizes of relationships between predictors and CORE-OM NR score. *Note*. Forest plot of effect size, on the y axis, for relationship of predictor variables, on the x axis, and CORE-OM NR score. Effect sizes are eta squared values with parametric 95% CIs as vertical error bars around the observed values. For predictors other than gender and age, simple effects with no other predictors are shown (“Raw”) and then the simple effects after partialling out effects of gender, age and of both gender and age are shown. The raw effect of gender is show with the effect after partialling out the effect of age, and vice versa.

**Table 1 ijerph-17-08520-t001:** Survey’s variables and open-ended question description.

Dependent Variable	Type	Description
NR scale from CORE-OM	Ordinal five-point item scale	28 items assessing psychological distress
Questionnaire’s variables	Type	Description
Non COVID-19 Demographics		
Sex	Categorical	Female, Male, Transgender
Age	Continuous	Years of age
Marital/Relationship Status	Categorical	Single, Married, Divorced, Other
Household size	Integer	Number of people in household
Children	Integer	Number of children
Occupation	Categorical	Technical-administrative, Teaching, Both
Hours of work	Continuous	Number of hours of work per week
COVID-19 Demographics		
Remote work	Dichotomy	Currently in remote work? (No/Yes)
Social isolation1	Dichotomy	Fulfilling the recommendations of social isolation to contain the contamination? (No/Yes)
Social isolation2	Continuous	Days in social isolation, going out only when necessary
Essential work	Dichotomy	Member of an essential work group? (No/Yes)
Other Essential	Dichotomy	Have a member of the household in an essential work group? (No/Yes)
Risk group	Dichotomy	Is at high risk if infected (in a risk group?) (No/Yes)
Infected self	Categorical	Have been infected with COVID-19? (No/Yes, mild symptoms, but not tested/Yes, moderate symptoms, but tested/Yes, tested)
Other infected	Categorical	Have other in household been infected? (No/Yes, mild symptoms, but not tested/Yes, moderate symptoms, but tested/Yes, tested)
Self-care/health activities		
Alcohol	Ordinal	Days of alcohol consumed per week (Never/1 or 2 days/3 or 4 days/Everyday)
Unprocessed food	Ordinal	Days of unprocessed food consumed pe week (Never/1 or 2 days/3 or 4 days/Everyday)
Exercising	Ordinal	Days of physical exercising per week (Never/ 1 or 2 days/3 or 4 days/Everyday)
Relax activity	Ordinal	Days of having relax activities (e.g., meditating) per week (Never/1 or 2 days/3 or 4 days/Everyday)
General and professional support		
Others help	Ordinal	Perception of help being available (Never/Rarely/Sometimes/Very often/Almost always)
Listeners avaliable	Ordinal	Perception of having listeners available (Never/Rarely/Sometimes/Very often/Almost always)
Psychological support	Ordinal	Experience of counselling (Never/In the past/Currently)
Psychiatric support	Ordinal	Experience of psychiatric support Never/ In the past/Currently)
Qualitative Question	Type	Description
Major concerns	Qualitative, open-ended	Subjective narrative of current major concerns related to pandemic

**Table 2 ijerph-17-08520-t002:** Major concerns: CQR domains and categories.

Domain/Categories	%	Illustrative Quotes
Work		
Work overload	45.94	*My workload has increased exponentially.*
Hyper connectivity and digital fatigue	22.52	*Having to work long hours on the computer is exhausting. Online teaching eliminates non-verbal language.*
Pressure from managers	9.91	*We are doing everything possible and impossible to deliver everything that is requested.*
Difficulty in establishing limits and routines	6.31	*Work invading holidays and weekends*
Concern and problems with productivity	9.01	*I am concerned that I am not being as productive as in person.*
Lack of access to tools and conditions for work	7.21	*Lack of facilities for work.*
Health		
Fear of contagion	44.05	*I am afraid to infect myself and parents who are at risk group.*
Symptoms and complains	25	*Sleep impairment, worsening diet and weight gain.*
Concerns with family members	17.86	*My father being hospitalised in serious condition.*
Restrictions of self-care activities	13.09	*Reduced physical exercise.*
Isolation		
Longing and loneliness	56.16	*Living alone at times brings the feeling of loneliness*
Lack of Freedom	43.84	*It bothers me not being able to leave the house, not being able to carry out my tasks with freedom and autonomy*
Personal life and routine		
Reconcile multiple tasks	53.33	*Setting boundaries between personal time and working time.*
Children care	18.90	*Homeschooling*
Housework	13.33	*Another stressful thing is the overload of household chores (food routine, cleaning activities, housekeeping).*
Lack of personal time	2.22	*The lack of time for myself, even to do nothing.*
Family conflicts	8.89	*Lack of dialogue and the mobile phone that is always in everyone’s hands!*
Difficulties with routine	3.33	*It is difficult to adapt to homeworking and find routines with any pleasure in them at home, with a routine that adds pleasure to doing it.*
Social environment		
Denial of COVID-19 severity	20.83	*To hear people saying with conviction that COVID-19 is just a ‘little flu’, both on television and on the social network*.
Political and economic insecurity	37.50	*General insecurity in the management of the crisis in Brazil*
Social impact of pandemic	12.50	*In general, the concern with society, with the conditions in which other people are living*
Negative news	29.17	*Negative highlights that are given by some media referring to the numbers of people infected with COVID, they should highlight the people who are cured.*
Future		
Prospects of losing jobs and income	58.21	*The risk of losing my job (in case the crisis gets worse) and financial support in the near future also leaves me in an uncomfortable situation*
Uncertainty about returning to normal	41.79	*The feeling of uncertainty, of deadline without limits*

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
