# Peer review of "Psychological Distress of University Workers during COVID-19 Pandemic in Brazil"

_ijerph, 2020, doi:10.3390/ijerph17228520_

Round 1

Reviewer 1 Report

Psychological distress during COVID-19 pandemic distancing precautions in Brazil

Comments to the authors

Thank you for the opportunity to review this manuscript. The work presented here is important and compelling. Overall, the paper is well organized, however, there are some concerns about the manuscript in its current form that I would like to see addressed. Here are some comments that I hope will be helpful to the authors and future readers of the paper if published.

Title: The title may also require some modification to reflect the actual study. The use of the word distancing precaution is ambiguous and leaves me wondering what exactly the study is about. Is it referring to social distancing, work from home, closure of schools and business etc.…

Abstract – could also benefit from specific results including the 95% CI values

Introduction

The introduction is generally well written and establishes some bits and pieces of the background and the justification of the study. However, it may require some refocusing so that it establishes a stronger beginning for the reader. The authors do a good job in paragraphs 1 & 2 by establishing the psychological and the social effects of the pandemic which is the focus of the study. The authors then give examples of studies in China that have revealed the psychological suffering due to the pandemic. In line 45, you indicate that young adults are the most affected group… although I think this is not the target population in the current study. As a reader, I get lot at this point as I expect that in the subsequent sections, I should be seeing this argument built further. But what I see in the subsequent paragraph, line 50-51 is that children and health professionals are also most affected. At this point, there are several points put across but with limited information to keep me as the reader engaged and to allow me know exactly who the population of focus is in the paper. In short, the introduction might benefit more from some literature that gives justification for the current target population – university employees.

Also, in this section, the authors might consider defining the key concepts used in the paper so as to provide an opportunity for the reader to have a clear take off diving into the other sections of the paper. For example, the operational definition of psychological distress and other predictor variables.

Materials and methods

The methods section also requires some beefing up to enable the reader to understand how the study was conducted and for replicability. Line 89-90 – what is the name of the university from where the data was collected? How was the sample of 407 arrived at? Did you target the entire population (N=1850) employees and got a response from 407? Is this sample representative enough to allow for generalization even if it is at the university of study? The sampling method might have had an impact on the gender distribution. By this, my question is why more female (68%) than the male? Out of the total population, 1850, what is the gender distribution? These could be some of the limitations to the study that we could be upfront about.

Line 92 – the use of M and Md to represent sample mean and median seems not standard. You may consider using what is familiar like X-bar Ⴟ etc

In section 2.3 – Instrument

This section might benefit more if it focuses on measurement of the variables. For example, in line 107, there is the mention of non-risk (NR) score which is considered the main response variable in the paper. As a reader, it is unclear to me how this is measured and why this should be the main response variable.

In line 113, measure of internal consistency (Cronbach’s ….) is given for a study in the UK but I am not sure how this relates to the current study. If the internal consistency of the current study was also measured, it will be great to give it either at this point or in the results section to allow for comparison or even to measure validity and reliability of the current study which I see in section 3.4

Throughout the paper, there is the use of simple, medium, and complex effects. As a reader, I do not see anywhere a clear definition of what is meant by these concepts is given. In this section, it might be great to give a clear distinction of how these concepts are used in the paper, especially in the results section. E.g. In line 259… “there was a complex effect of gender and household size….” I do not know what the authors mean by this

Section 2.4 – Procedures

Requires more detail on the implementation of the study… Line 127 – why was the form open for ten days? Rationale….

Section 2.5 – Data analysis

The use of CQR in analyzing the open-ended questions is interesting and is a great way to capture the divergent views that emerge from the survey respondents. Some points of clarity – it will be great to highlight the questions that were open-ended and the key objectives for these.

In line 140 – “First pairs of independent raters examined 50 answers each …..” and we had 6 raters, an just curious to understand how this was conducted and how many responses (n=407) did the raters each do…?

Line 158 … Whenever possible, 95%CI … are used – It is unclear to me what determines the possibility in this case. I imagine that based on the analysis performed and the marked output, CIs can be generated

Section 2.6 Ethical consideration

Other issues of ethics that were observed should be highlighted

Results

Line 214 – N=487? It might be typo so please check and confirm

Section 3.4.1 Association…

I imagine that the point of interest in this section is to determine association between the outcome and predictor variables while controlling for other predictors. The title of this section is rather confusing in that it seems to me we are determining association between predictor variables. Furthermore, the measure of association is not clearly presented in this section. The results here are presented in %s which is kinder confusing. I believe there are some specific measures of association that could be used to present the results in a more interesting manner.

Figure 1, 2 and 3, the keys are in Spanish – change to English for consistency

Figure 1 – has the sub-category – transgender – this was just a single individual. Wondering if it would be logical to drop this as 1 respondent is not sufficient to ….

In line 259… “there was a complex effect of gender and household size….” I do not know what the authors mean by a complex effect (mentioned above already)

Section 3.4.2 Notable…

The authors present the results on association in form of η2 and M as the measure of association accompanied by 95% CI. In my view, I find this a bit confusing and somehow inaccessible to the reader. This is because, the interpretation of the results might not be straight forward. For example, in line 273 days of alcohol consumed (η2 = 0.022; 95%CI [0.01-0.046])…., I am not sure whether this means that alcohol consumption is protective and reduces the effect of COVID or vise versa… just an example

 The use of mean (M) from line 282 and the accompanying CI is also difficult to interpret in my own view. M in this case it refers to the actual mean or the mean differences and why use the M here and not the η2.

Discussion & conclusion

This section is well written as it clarifies some of the concerns and the conclusions are drawn from the research findings. However, a clear presentation of the results will strengthen the discussion even more as it will allow the reader to engage more with the findings while seeing how the findings relate to other pieces of work in the same topic.

There is some editorial work required to allow for clarity and flow of ideas in this section. E.g. Line 317 – anyone can be infected….  

Acknowledging the study limitations in this section is also important to enable the reader know some of the precautions to consider while interpreting the findings.

Reviewer 2 Report

The authors aim to describe the psychological distress of a specific cohort of workers during COVID-19 pandemic in Brazil and describe the associated conditions. The subject and the aims of the papers are interesting and useful to better understand the psychological impact of the pandemic in Brazil. The methodology and the design used appears good, although the generalizability of results is a concern. Moreover, in my opinion, major revisions have to be made in particular in improving the clarity of the paper and focusing on the main results and conclusions. Indeed, many information has been collected through the survey and several analyses have been conducted, both qualitative and quantitative, so that in my opinion the reader might find difficult to identify the core results and their implication to develop psychological intervention (the primary goal of the study).

Here are the main concerns:

  • In the background, many results on the pandemic in Brazil have been discussed, however as the journal has an international focus, authors should report some references on similar studies at the international level, including studies on non-health workers.
  • In the background and/or in the discussion, authors should explain the reason for considering the psychological distress of university workers and/or what their characteristics can be that make them particularly vulnerable. In the discussion, considerations should be added on the generalizability of the results to another context of non-health workers and a comparison with other non-health workers or general population in the country should be done (if possible).
  • As regards method and results, I suggest adding the description of the variables in the survey (apart CORE-OM) and results of the univariate analysis in the Appendix and, in the text, I would add a table with a description of the main variables to describe the sample. For example, you reported the percentage of technical-administrative workers, which other professionals were included? This might add clarity and transparency on the collected predictor variables. Moreover, the exploratory aims of the analysis declared by the authors in the methods are clear, but I suggest to organize the results reporting clearly the main ones. For example, in the paragraph 3.4.2 authors should synthesize clearly the main relationships emerged and reporting data in the table, deleting data from the text.
  • In the discussion, to improve its readiness, authors should sum up the main findings and add a discussion on the practical implication for psychological interventions.
  • In the discussion, the authors mentioned the similarity between their results and other studies on age and gender. Considering the relevance of age and gender for the prevalence of common mental disorders, I suggest adding other studies to compare the results of the current paper and discuss the results more in-depth, also considering that gender resulted associated to variable ‘support from a psychologist’ in section 3.4.1.
  • In the discussion, a section with the study’s limitations and strengths is lacking and should be added. In particular, I would suggest a section on the generalizability and representativeness of the results. As regards the response rate, it has been not explicitly reported, although if I understand correctly it is low with participants being 407 over 1850.
  • In the discussion, the result on CORE-OM has not been discussed. Is it possible to compare the data on psychological distress with other similar population?
  • As the characteristics and the timeframe of restrictive measures and the spread of the disease are different in each country, more information on the situation at the time of the survey should be added to improve the understanding of stressors and distress. Some considerations on whether the data reported here can be representative of the whole Brazilian situation should be added.

Some minor concerns

  • Figures have some words non in English; Figure 1 is not clear without colour; Figure 4 needs a legend.
  • The sentence at line 226 is not clear.

Round 2

Reviewer 2 Report

The manuscript has been significantly improved by authors and authors replied to my concerns. I only have noticed some typos in the added text that should be corrected and I suggest minor spell check.